# ADAPTIVE TEST-TIME INTERVENTION FOR CONCEPT BOTTLENECK MODELS

**Matthew Shen**[*]
Department of Statistics
Columbia University
ms7079@columbia.edu

**Aliyah R. Hsu**[*]
Department of EECS
UC Berkeley
aliyahhsu@berkeley.edu

**Abhineet Agarwal**
Department of Statistics
UC Berkeley
aa3797@berkeley.edu

**Bin Yu**
Department of Statistics, EECS
Center for Computational Biology
UC Berkeley

## ABSTRACT

Concept bottleneck models (CBM) aim to improve model interpretability by predicting human level "concepts" in a bottleneck within a deep learning model architecture. However, how the predicted concepts are used in predicting the target still either remains black-box or is simplified to maintain interpretability at the cost of prediction performance. We propose to use Fast Interpretable Greedy Sum-Trees (FIGS) to obtain Binary Distillation (BD). This new method, called FIGS-BD, distills a binary-augmented concept-to-target portion of the CBM into an interpretable tree-based model, while maintaining the competitive prediction performance of the CBM teacher. FIGS-BD can be used in downstream tasks to explain and decompose CBM predictions into interpretable binary-concept-interaction attributions and guide adaptive test-time intervention. Across 4 datasets, we demonstrate that our adaptive test-time intervention identifies key concepts that significantly improve performance for realistic human-in-the-loop settings that only allow for limited concept interventions. All code is made available on Github. [1]

## 1 INTRODUCTION

Deep learning (DL) has achieved impressive performance in various domains such as computer vision (CV) and natural language processing (NLP). Despite their success, DL models are often uninterpretable. Concept bottleneck models (CBMs) (Koh et al., 2020) aim to improve the interpretability of DL models by explaining predictions in terms of human-understandable "concepts". CBMs can functionally be decomposed into two models: an input-to-concept model and a concept-to-target (CTT) model. Prior CBM work typically uses a linear CTT model for interpretability (Koh et al., 2020; Wong et al., 2021; Ludan et al., 2024). This limits the expressivity of the overall CBM, hurting downstream performance which instead requires CTT models that can capture more complex relationships between concepts. CBMs, especially with practitioner intervention (i.e., check correctness and edit prediction if necessary), have the potential to improve the trustworthiness and usability of models for cases like medical diagnosis (Oikarinen et al., 2023; Yuksekgonul et al., 2023). However, current concept intervention work does not account for difficulties of interventions in high pressure environments with practitioners lacking full domain experience: a surprisingly common scenario where machine learning could be most effectively utilized.

In this work, we address the lack of CTT model interpretability in all concept settings, especially when using complex models to capture complicated CTT relationships (i.e. NLP). We propose the distillation of the CTT portion of the CBM (CTT CBM) with an interpretable Fast Greedy

---

[1]https://github.com/mattyshen/adaptiveTTI
[*]Equal contribution.

Sum-Trees (FIGS) model (Tan et al., 2025). This allows human practitioners to understand the predictions made by the CTT CBM through a sum of contributions depending on interactions of concepts. The FIGS model, through it's construction, also adaptively proposes and ranks concepts that are of highest priority for a practitioner to intervene on. The proposed distillation and adaptive test-time intervention process is visualized in Figure 1.

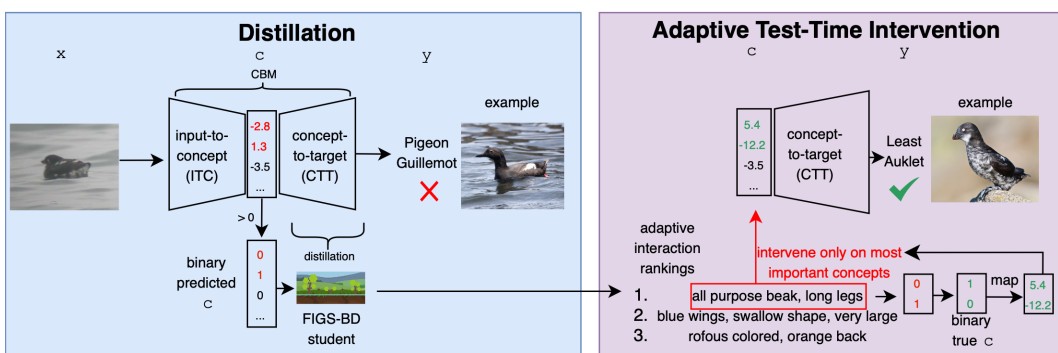

Figure 1: The CBM incorrectly identifies "long legs" in the image, perhaps due to the spurious correlations between water and long legged birds like seagulls. FIGS adaptive test-time intervention (ATTI) recommends a small number (2) of concepts based on a binarization of predicted concepts (including "long legs") to intervene on, which results in the correct prediction.

## 2 RELATED WORK

### 2.1 CONCEPT MODELS

To improve model interpretability, models can be bottlenecked on human-level "concepts," popularized by Koh et al. (2020). The usage of concepts to understand models has expanded to analyzing models post-hoc (Yuksekgonul et al., 2023), using other models (i.e. LLMs) or adapting models to iteratively generate and refine concepts for tasks (Oikarinen et al., 2023; Schrodi et al., 2024; Chen et al., 2019; Li et al., 2024; Ludan et al., 2024). Some concept models further learn soft rules (Vemuri et al., 2024) or (decision tree) structures (Nauta et al., 2021), using the predicted concepts to improve interpretability and practitioner usage. Xu et al. (2024) propose energy based CBMs to address limitations of CBMs in capturing nonlinear interactions, and similarly recognize the lack of a principled approach to test-time intervention.

### 2.2 KNOWLEDGE AND MODEL DISTILLATION

In knowledge and model distillation, introduced by Hinton et al. (2015), a compact student model is trained on the predictions of a larger, more complex teacher model to improve inference speed, computation, or even interpretability, while maintaining competitive predictive performance (Jiao et al., 2020). Having an interpretable model that mimics a complex model through distillation can increase the trustworthiness of complex models, streamlining their use into real-life environments.

## 3 FIGS BINARY DISTILLATION – FIGS-BD

We utilize the Fast Interpretable Greedy Sum-Trees (FIGS) algorithm (Tan et al., 2025) to distill the CTT CBMs. We modify the original FIGS algorithm by restricting the maximum depth of the trees learned to maintain interpretability and introduce a multi-output variant to distill the soft-labels (i.e. target logits or probabilities) of CTT CBMs. The FIGS composition of a flexible, yet upper bounded, number of trees and "rules" is inherently interpretable, and practitioners can thus understand predictions made by the (CTT) CBM (and the FIGS student model) as a sum of interactions between concepts. In traditional CBMs, predicted concepts are often logits, which are highly uninterpretable and bring about unnecessary uncertainty to practitioners. An "on" or "off"

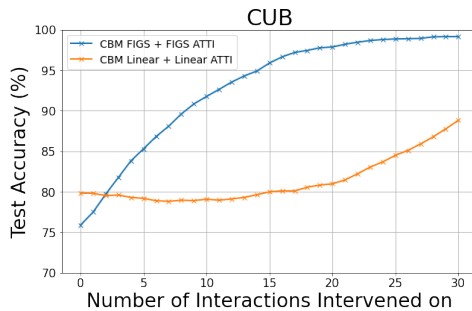 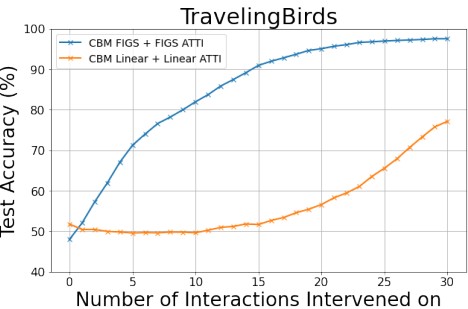

Figure 2: Effectiveness of adaptive test-time interventions for different concept-to-target models. Note the $x$-axis enumerates the number of *interactions* (of at most 3 concepts) intervened on.

binary representations of concepts alleviates this uncertainty and lack of interpretability. Thus, we binarize CBM predicted concepts with data-driven (minimize distance between true concepts) or interpretable ($> 0$) thresholds, and distill the CTT CBM using these binary concepts, (teacher) predicted target logits, and FIGS, which we call FIGS-BD.

**Why FIGS?** Predicting targets from binary concepts constitutes learning a Boolean function $f : \{0, 1\}^d \to \mathbb{R}$. All Boolean functions can be expressed as Fourier series (Spiro, 2016). Learning this Fourier series exactly requires exponential samples and time; FIGS-BD instead greedily approximates $f$ by constructing a sum of shallow trees.

## 4 DATASETS AND TEACHER MODELS

Our experiments contain two tasks: CV and NLP. For CV, we train CBMs (Koh et al., 2020) on the Caltech-UCSD Birds-200-2011 (CUB) dataset (Wah et al., 2011) and the TravelingBirds (Koh et al., 2020) dataset, which is a variant of CUB where the image backgrounds associated with each bird class are changed from train to test time. CUB and TravelingBirds both pose as challenging prediction tasks with a high number of classes, while TravelingBirds also showcases a distribution shift from train to test time. For NLP, we train LLM-based Text Bottleneck Models (TBMs) [*] (Ludan et al., 2024) on the AGNews topic classification dataset (Zhang et al., 2015) and the CEBaB restaurant reviews (Abraham et al., 2022) dataset (regression task). These two datasets are deemed to have complicated concept interactions in nature that could not be captured in previous TBM work with a linear CTT model (Ludan et al., 2024). More details of experiments are in Appendix A.1.

## 5 DISTILLATION AND PREDICTION PERFORMANCE

Table 1 displays the best test performing CBM/TBM models (with CTT model specified), as well as FIGS and XGBoost (Chen & Guestrin, 2016) student models' test prediction performance on the CUB, TravelingBirds, AGNews, and CEBaB datasets. A complete table, with other comparative baselines (decision tree and random forest), is in Appendix A.3. Depending on the dataset, the relationship between concept and target can either be very simple or very complex. CUB and TravelingBirds have a fairly linear CTT relationship. For AGNews and CEBaB, complex Transformer models capture the CTT relationship the best, necessitating distillation to improve interpretability and prediction understanding. As evident in the small difference between teacher and student model prediction performance, FIGS-BD is distilling effectively, even in out-of-sample data. FIGS-BD achieves over 92.5 % of the performance of its teacher CBM on the test sets of all surveyed datasets, while generalizing better than the original CBM in some cases (CEBaB). FIGS-BD performs closely with XGBoost in the CUB and CEBaB datasets, and even outperforms XGBoost in the AGNews and TravelingBirds datasets despite being smaller (significantly less rules) and more interpretable (XGBoost fits a separate model for each class, while FIGS-BD fits a single multi-output model).

---

[*]Following the definition in Section 1, a TBM is also a CBM. However we refer to the models used in the NLP tasks as TBMs in the following sections to differentiate from the CBMs used in the CV tasks.

Table 1: Best CBM/TBM test prediction performance with FIGS-BD and XGBoost student models across the 4 datasets. "Teacher Pred" and "Student Pred" denote teacher and student test prediction performance, respectively.

| Dataset | Teacher | Student | Teacher Pred | Student Pred |
|---|---|---|---|---|
| CUB (Acc %) | CBM Linear | FIGS-BD | 79.8 | 75.9 |
| | - | XGBoost | - | 75.9 |
| TravelingBirds (Acc %) | CBM Linear | FIGS-BD | 51.8 | 47.9 |
| | - | XGBoost | - | 47.7 |
| AGNews (Acc %) | TBM Transformer | FIGS-BD | 89.6 | 88.8 |
| | - | XGBoost | - | 88.0 |
| CEBaB (R-squared) | TBM Transformer | FIGS-BD | 0.868 | 0.871 |
| | - | XGBoost | - | 0.877 |

# 6 ADAPTIVE TEST-TIME INTERVENTION USING FIGS-BD

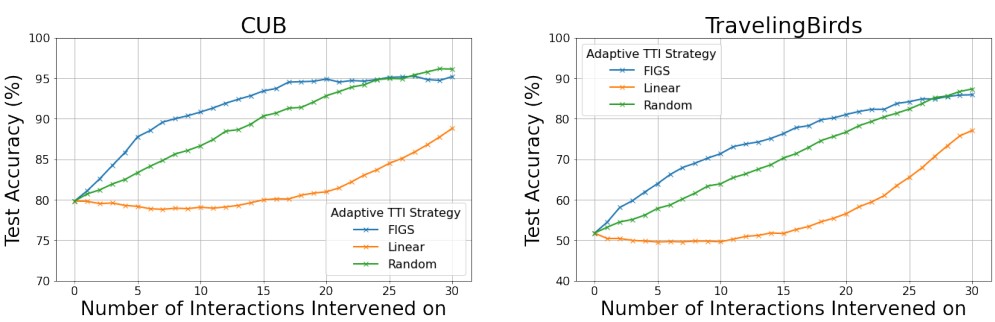

Figure 3: Performance of CBM linear with adaptive test-time interventions for concepts suggested by different CTT models. FIGS ATTI greatly out-performs Linear ATTI.

In high-stakes environments (e.g., emergency rooms), practitioners cannot intervene across all concepts but rather can only do so for a limited number of concepts. In such scenarios, identifying an important ranking of concepts is crucial for accurate prediction. In this subsection, we consider the task: *adaptive test-time intervention (ATTI)* in which a human is allowed to intervene on a small number of concepts for a given test-example. We show how FIGS-BD can be used to adaptively rank the most important concepts for a human to validate before prediction.

We propose constructing a sample-specific ranking of concepts based on the highest variance of absolute predictions (across the target dimension) path, from where the concepts are identified, that the sample falls down. Algorithm 1 describes this process in pseudo code. Similarly, for linear CTT portions, we propose ranking concepts based on the highest variation of absolute values of the product of fitted coefficients and predicted concept values. We believe that higher variance (across the target dimension) represents "volatile" contributions that are the most important to intervene on. More details can be found in Appendix A.2.

**Quantitative prediction improvement on CUB and TravelingBirds**  We conduct an experiment where a practitioner is allowed to intervene on the top$-k$ interactions of concepts for a test sample recommended by various TTI methods. We consider top concepts recommended by FIGS-BD, a linear CTT, as well as random selection. We plot the results in Figure 3. FIGS identifies concepts that are much more relevant for making a correct prediction, indicating its utility in identifying relevant concepts for humans to validate.

Additionally, we conduct an ablation study comparing the original linear CTT model (CBM Linear) with linear ATTIs and the FIGS-BD CTT model (CBM FIGS) with FIGS ATTIs. We plot the results in Figure 2. The FIGS-BD CTT model quickly surpasses the linear with a practitioner's interventions, reaching drastically higher test accuracy %s with a moderate to large number of interventions. Specifically, in as few as 3 and 1 interaction interventions for CUB and TravelingBirds,

respectively, the FIGS-BD CTTs outperforms linear CTTs. This highlights the impact of editing with binary values (rather than with predicted training data quantiles) and the effectiveness of FIGS ATTI. On TravelingBirds, FIGS-BD requires far less interventions (7) to reach the same accuracy as the maximum intervened on (30) linear model, potentially further disentangling the detrimental spurious correlation that was propagated into the original linear CTT model.

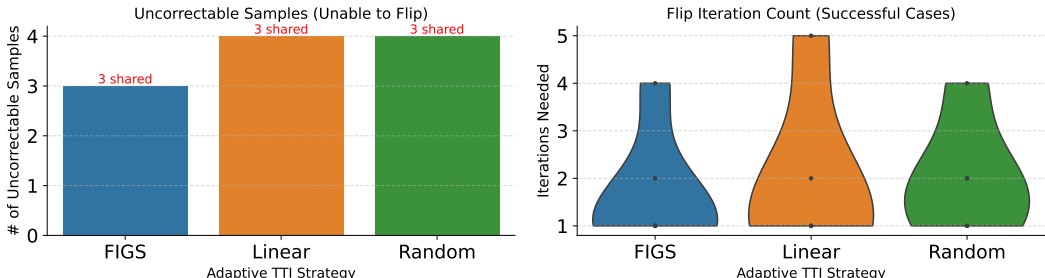

Figure 4: **Left:** number of uncorrectable samples of each intervention method. **Right:** count of iterations of intervention needed of each method to flip a wrong prediction into a correct one.

**Effectiveness for correcting model prediction on AGNews and CEBaB** Unlike CUB and TravelingBirds, AGNews and CEBaB lack human-labeled concepts. To address this, we manually annotate a small set of misclassified samples from these datasets with concept labels [*]. We then evaluate different intervention methods by measuring how many iterations of interventions it takes to "flip" an incorrect prediction to a correct one, intervening sequentially via ATTI interactions rankings.

Figure 4 shows results on AGNews and CEBaB (combined) using the linear CTT model with various ATTI strategies. FIGS ATTI consistently achieves successful flips with fewer iterations compared to other ATTIs. Moreover, it results in fewer uncorrectable samples (i.e., samples for which all recommended interventions fail to correct the prediction), and its uncorrectable samples are a strict subset of those from other methods, indicating that it successfully recovers some cases others cannot. In contrast, linear ATTI even underperforms random ATTI, suggesting that linear models struggle to recommend reliable concepts for intervention.

As a case study, we highlight an example from CEBaB where FIGS ATTI was the *only* method able to correct the model's incorrect prediction, requiring just *one* intervention. The linear CTT model initially misclassified the review `"My dining experience was one of the best. The food and service was outstanding. Everyone was very friendly just could have turned down the volume of the music a little."` with a rating of 5 instead of the ground-truth rating of 4. The model overemphasized the concept `"Customer Expectations"`, which was not present in the review. FIGS ATTI correctly identified `"Customer Expectations"`, `"Overall Satisfaction"`, and `"Service Quality"` as the most critical concepts for intervention, enabling a human to downweight the erroneous concept and produce the correct rating. In contrast, the other methods failed to surface this issue in their recommended interactions. In short, FIGS-BD ATTI's superior performance results from its ability to identify concept interactions are crucial in determining between two competing classes effectively for human intervention.

## 7 CONCLUSION

In this paper, we propose FIGS-BD: an algorithm to distill binary-augmented concept-to-target portions of CBMs to interpret their predictions as contributions of concept interactions. From the FIGS-BD student model, we introduce adaptive test-time introduction, which requires CBMs to propose a small number of concepts to be validated before prediction. FIGS-BD identifies more relevant concepts for accurate prediction. Future work involves extension to post-hoc CBMs, further empirical evaluation, and counterfactual predictions with FIGS.

---

[*]Annotations were performed by three PhD students specializing in statistics or computer science.

ACKNOWLEDGEMENTS

We gratefully acknowledge partial support from NSF grants DMS-2209975 and DMS-2413265, NSF grant 2023505 on Collaborative Research: Foundations of Data Science Institute (FODSI), the NSF and the Simons Foundation for the Collaboration on the Theoretical Foundations of Deep Learning through awards DMS-2031883 and 814639, NSF grant MC2378 to the Institute for Artificial CyberThreat Intelligence and OperatioN (ACTION), NIH grant R01GM152718 (DMS/NIGMS), a Berkeley Deep Drive (BDD) Grant from BAIR and a Dean's fund from CoE, at UC Berkeley.

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

## A  APPENDIX

### A.1  MODEL ARCHITECTURES AND HYPERPARAMETERS

For all datasets and CBM/TBM models, we utilize code and the majority of architectures provided by the authors of respective papers. For more complex concept-to-target (CTT) portions of the CBM, we modified provided code and scripts to train and evaluate the model. For FIGS models, we have contributed to the `imodels` (Singh et al., 2021) package (specifically, the FIGS implementation) to restrict the maximum depth of trees, handle multi-output prediction tasks, and create cross validation (CV) models. We have then adapted that model for ATTIs.

### A.1.1  CUB AND TRAVELING BIRDS

The Caltech-UCSD Birds-200-2011 (CUB) dataset (Wah et al., 2011) and the TravelingBirds (Koh et al., 2020) dataset contain $n = 11,788$ photos of birds with 200 bird class labels. Every observation in the dataset comes with human-labelled annotations regarding concepts present in the image, which facilitates our ATTI experiments. We reduce the number of concepts used in the same procedure as described in Koh et al. (2020). We utilize the code, instructions, and some trained models provided by Koh et al. (2020). We modify parts of their Github repository to incorporate more complex concept-to-target models. Specifically, we include MLP with 1 hidden layer, MLP with 2 hidden layers, and a simple Transformer model (encoder-only). All MLPs have the same hidden size, set to be 250 for CUB and TravelingBirds. The Transformer model utilizes multi-headed attention (Vaswani et al., 2023) with 4 heads, a MLP with 1 hidden layer of hidden size 250, and then a linear classifier layer. For the input-to-concept portion of the CBM, we utilize the Inception V3 (Szegedy et al., 2015) model, and for the overall model, utilize the overall Joint training process with $\lambda = 0.01$. All hyperparameters regarding training are the same as in Koh et al. (2020). Due to the complicated 200 class prediction task posed by CUB and TravelingBirds, we utilize a FIGS CV model to determine the hyperparameters that result in the strongest FIGS-BD model. We use an interpretable rule of $> 0$ (a positive concept prediction results in 1, negative results in 0) to binarize concept features before FIGS distillation. We search over [125, 200] rules, [30, 40] trees, and [3, 4] max depth. For CV results in Table 1, the post cross-validation fitted FIGS-BD model results in 200 rules, 30 trees, and max depth of 3 for both CUB and TravelingBirds.

### A.1.2 AGNews and CEBaB

AGNews contains $n = 7,600$ news articles and 4 class labels (world, sports, business, and sci/tech) for news topic classification. CEBaB contains $n = 1,713$ restaurant views and their corresponding ratings (1-5) from customers as labels, and we formulate it as a regression task. For both datasets, we randomly split them into $n = 1,500$ train set and $n = 250$ test set for training and evaluation. To be comparable to the original TBM (Ludan et al., 2024) paper, we use GPT-4 (`GPT-4-0613`) (OpenAI et al., 2024) as the underlying LLM for concept generation and concept measurement in the input-to-concept portion of the TBMs. The original TBM code uses Scikit-learn (Pedregosa et al., 2011) for training linear regression (regression task) and logistic regression (classification task) for the concept-to-target portions of the TBMs. We modify parts of their code to incorporate more complex concept-to-target models. Specifically we include MLP with 1 hidden layer, MLP with 2 hidden layers, and a simple Transformer model (encoder-only). All MLPs have the same size set to be 50 for both AGNews and CEBaB datasets. The Transformer model utilizes two blocks of multi-headed attention (4 heads) + MLP with 1 hidden layer (hidden size 52) module, and then a linear classifier layer. All hyperparameters regarding training are the same as in Ludan et al. (2024), except for the refinement trial size, which we set to be 500 for training the more complicated CTT models (MLPs and the simple transformer). We use one-hot-encoding to binarize concept features before FIGS distillation. We search over [100, 200, 250] rules, [20, 30, 50] trees, and [3, 4] max depth. For the NLP results in Table 1, the post cross-validation fitted FIGS-BD model results in 154 rules, 50 trees, and max depth of 3 for both CEBaB and AGNews.

## A.2 Adaptive TTI Interventions

---

**Algorithm 1** FIGS-BD ATTI algorithm

---

1: **FIGSBD_ATTI**($f_{\text{FIGS}}$: FIGS-BD model, x: $\mathbb{R}^{n_{\text{concepts}}}$)
2: $all\_trees = trees(f_{\text{FIGS}})$
3: $tree\_predictions = [\,]$
4: $tree\_paths = [\,]$
5: **for** $tree$ in $all\_trees$ **do**
6: $\quad tree\_prediction.\text{append}(tree.\text{predict}(x))$
7: $\quad tree\_paths.\text{append}(\text{path}_{tree}(x))$
8: **end for**
9: $predictions\_and\_paths = \text{zip}(tree\_predictions, tree\_paths)$
10: $rankings = sort(predictions\_and\_paths, \text{lambda } x_{pred} : \text{var}(|x_{pred}|) \text{ or max}(|x_{pred}|)))$
11: **return** $rankings$

---

FIGS-BD ranks interactions of concepts that are embedded in the structure of its collection of trees. The ranking procedure is described in pseudo code in Algorithm 1. Thus, every set or interaction of concepts to be intervened on in Figure 3 are of size maximum depth of grown tree. Note that this is not always equal to the maximum depth hyperparameter of the model, as the FIGS model does not have to grow to full depth. Additionally, concepts are re-used in some learned interactions, so intervention is not as effective after many interventions have occurred (and are thus the most impactful for the earlier sets of interactions intervened on). For each observation, these interactions of varying size are ranked based on a heuristic function (variance of absolute value of multi-output prediction and maximum of absolute value for 1-dimensional output prediction). For random ATTI, we randomly choose concepts without replacement and group/parse them of corresponding size to every FIGS ATTI to make them comparable to FIGS ATTI. For linear ATTI, we rank the $n_{\text{concepts}}$ concepts based on variance of absolute value of product of concept prediction and concept coefficient, and group/parse them of corresponding size to every FIGS ATTI to make them comparable to FIGS ATTI. Note that when talking about variance, we refer to the variance of predictions across the multi-output target dimension.

For all of our experiments, the best FIGS-BD student models grow to depths of 3, which means that the maximum size of every interaction or cluster of concepts intervened on is 3.

For CUB and TravelingBirds, as done by Koh et al. (2020) for interventions, we replace the predicted concept values with the 5th quantile and 95th quantile of the predicted concept in the

training data if the true concept is 0 and 1 for the original CBM (linear CTT), respectively. This is denoted as "map" in Figure 1. This can result in prediction performance degrading as replacing predicted values for a specific instance with training values, even if the pre-intervention and post-intervention concept values agree in some way (one could perhaps argue for equivalence in sign meaning an agreement, but there is no exact way without uncertainty to determine if a CBM predicted a concept correctly).

## A.3 FULL CBM AND STUDENT MODEL PREDICTION AND DISTILLATION RESULTS

Table 2 contains all teacher and student test prediction performances across a variety of teacher models and selection of student (regression) models: FIGS, XGBoost (Chen & Guestrin, 2016), Random Forest (RF) (Breiman, 2001), and Decision Tree (DT) (Breiman et al., 1984). The teacher models vary in their concept-to-target portion, in which we consider Linear, MLP1, MLP2, and Transformer concept-to-target models. Note that FIGS-BD was trained using cross-validation, meaning that it is likely that if the teacher model is more complex, the FIGS-BD student model consists of more trees, more rules, and more depth. For CUB and TravelingBirds, we restricted XGBoost and RF to 30 trees (the same amount as the cross-validation-chosen FIGS-BD model). We depth-restricted XGBoost, RF, and DT to 3 (same as cross-validation chosen FIGS-BD model), 7 or 8, and 7 or 8, respectively, and chose the best performing model. We choose depth or 7 or 8 because there are 200 classes in the CUB and TravelingBird tasks, so we need enough expressivity (and leaf nodes: $2^7 = 128, 2^8 = 256$) to achieve strong performance. For AGNews and CEBaB, we restricted XGBoost and RF to 50 trees, and depth-restricted XGBoost, RF, and DT to 3 (same as cross-validation chosen FIGS-BD model), 2 or 3, and 2 or 3, respectively, following the same logic. The results displayed consist of XGBoost, RF, and DT of depth 3, 8, and 8, respectively for CUB and TravelingBirds, and of depth 3 for all three models for AGNews and CEBaB.

On all datasets, XGBoost displays strong performance, but we note that XGBoost was not restricted in terms of number of rules (only restricted in depth and tree) and XGBoost also grows a separate estimator per class/task, for example, resulting in $30 \cdot 200 = 6000$ total trees (for CUB and TravelingBirds) with max depth 3. Thus, XGBoost is highly uninterpretable and grows highly inefficient and dense trees. On the other hand, FIGS-BD grows sparser and is a number-of-rules restricted model, consisting of only 30 trees of max depth 3 for CUB and TravelingBirds. RF and DT perform significantly worse than XGBoost and FIGS-BD, while RF is also highly uninterpretable.

Table 2: Full CBM (teacher model) and student model test prediction performance across the datasets. "Teacher Pred" and "Student Pred" denote teacher and student test prediction performance, respectively. Top prediction performance for each dataset and model role (teacher or student) in bold. The second-best student performance is underlined. RF and DT denote Random Forest and Decision Tree, respectively.

| Dataset | Teacher | Student | Teacher Pred | Student Pred |
|---|---|---|---|---|
| | CBM Linear | FIGS-BD | **79.8** | **75.9** |
| | - | XGBoost | - | **75.9** |
| | - | RF | - | 64.4 |
| | - | DT | - | 50.2 |
| | CBM MLP1 | FIGS-BD | 79.0 | 73.7 |
| | - | XGBoost | - | **74.1** |
| | - | RF | - | 65.3 |
| | - | DT | - | 51.8 |
| CUB (Acc %) | CBM MLP2 | FIGS-BD | 78.0 | 72.7 |
| | - | XGBoost | - | **74.2** |
| | - | RF | - | 65.4 |
| | - | DT | - | 48.0 |
| | CBM Transformer | FIGS-BD | 77.4 | 72.2 |
| | - | XGBoost | - | **73.0** |
| | - | RF | - | 66.2 |
| | - | DT | - | 51.5 |
| | CBM Linear | FIGS-BD | **51.8** | **47.9** |
| | - | XGBoost | - | 47.7 |
| | - | RF | - | 38.4 |
| | - | DT | - | 28.5 |
| | CBM MLP1 | FIGS-BD | 49.2 | 48.5 |
| | - | XGBoost | - | **50.1** |
| | - | RF | - | 41.5 |
| | - | DT | - | 31.5 |
| TravelingBirds (Acc %) | CBM MLP2 | FIGS-BD | 49.6 | 49.1 |
| | - | XGBoost | - | **49.7** |
| | - | RF | - | 42.0 |
| | - | DT | - | 33.7 |
| | CBM Transformer | FIGS-BD | 47.5 | 47.1 |
| | - | XGBoost | - | **47.2** |
| | - | RF | - | 43.4 |
| | - | DT | - | 32.2 |
| | TBM Linear | FIGS-BD | 84.8 | 83.2 |
| | - | XGBoost | - | **86.8** |
| | - | RF | - | 82.8 |
| | - | DT | - | 81.2 |
| | TBM MLP1 | FIGS-BD | 84.4 | 80.8 |
| | - | XGBoost | - | **83.6** |
| | - | RF | - | 76.4 |
| | - | DT | - | 78.8 |
| AGNews (Acc %) | TBM MLP2 | FIGS-BD | 80.8 | 79.2 |
| | - | XGBoost | - | **82.0** |
| | - | RF | - | 76.8 |
| | - | DT | - | 76.8 |
| | TBM Transformer | FIGS-BD | **89.6** | **88.8** |
| | - | XGBoost | - | 88.0 |
| | - | RF | - | 83.2 |
| | - | DT | - | 83.2 |
| | TBM Linear | FIGS-BD | 0.761 | 0.797 |
| | - | XGBoost | - | **0.804** |
| | - | RF | - | 0.784 |
| | - | DT | - | 0.785 |
| | TBM MLP1 | FIGS-BD | 0.837 | 0.873 |
| | - | XGBoost | - | **0.882** |
| | - | RF | - | 0.864 |
| | - | DT | - | 0.863 |
| CEBaB (R-squared) | TBM MLP2 | FIGS-BD | 0.808 | **0.833** |
| | - | XGBoost | - | **0.833** |
| | - | RF | - | 0.813 |
| | - | DT | - | 0.812 |
| | TBM Transformer | FIGS-BD | **0.868** | 0.871 |
| | - | XGBoost | - | **0.877** |
| | - | RF | - | 0.847 |
| | - | DT | - | 0.786 |

