# OpenReview forum: "Adaptive Test-Time Intervention for Concept Bottleneck Models"
_ICLR.cc/2025/Workshop/BuildingTrust — BuildingTrust_

### Official Review · Reviewer_zH9b · 2025-02-24
**Review of "Enhancing CBMs Through Binary Distillation with Applications to Test-Time Intervention"**

**Rating:** 6
**Confidence:** 3

**Review:**

### Strength
This paper presents FIGS-BD, a novel approach to improving Concept Bottleneck Models (CBMs) by distilling their concept-to-target (CTT) relationships into an interpretable Fast Greedy Sum-Trees (FIGS) model. FIGS-BD improves interpretability without significantly sacrificing prediction accuracy, making CBMs more transparent for human-in-the-loop settings such as medical diagnosis. The paper also introduces adaptive test-time intervention (ATTI), which ranks critical concepts for human validation, improving model reliability. The method is evaluated on four datasets (CUB, TravelingBirds, AGNews, CEBaB) and shows that FIGS-BD achieves over 92.5% of the teacher model’s accuracy, while enhancing human interpretability and intervention effectiveness.

### Weakness
The paper lacks a detailed comparison against other interpretability approaches, such as post-hoc explainability methods or alternative model distillation techniques. Additionally, scalability concerns are not addressed—how well does FIGS-BD scale to larger models or real-world deployments? While the ATTI approach is promising, more rigorous human studies (e.g., involving real practitioners rather than simulated settings) would strengthen its impact. Finally, quantitative evaluation of interpretability is missing—how do human users perceive FIGS-BD explanations compared to traditional CBMs?

---

### Official Review · Reviewer_TJis · 2025-02-26
**Review of Enhancing CBMs Through Binary Distillation with Applications to Test-Time Intervention**

**Rating:** 5
**Confidence:** 3

**Review:**

## Strengths

- The paper addresses a practical limitation of CBMs: the challenge of maintaining interpretability in the concept-to-target portion while preserving model performance.
- The empirical results demonstrate that FIGS-BD achieves over 92.5% of the teacher model's performance across all datasets, suggesting minimal performance sacrifice for interpretability gains.
- The evaluation across both CV and NLP domains demonstrates the versatility of the approach.

## Weakness
- The paper does not offer a thorough theoretical comparison with other distillation or interpretable modeling methods for CBMs. Without this, it is hard to assess whether FIGS-BD offers significant theoretical advantages or if it is merely one of many viable approaches. This absence of contextualizing the contribution within the broader theoretical landscape weakens the paper's impact.
- While the paper references Fourier series representations of Boolean functions to justify binary representations, the analysis does not extend to formal guarantees or bounds on the approximation quality or interpretability trade-offs. It remains unclear how robust or generalizable this binary distillation is across different concept spaces.
- The method relies on distilling complex interactions into a set of shallow trees. However, there is no rigorous discussion on how the approach scales when the number of concepts increases. In high-dimensional settings, the combinatorial explosion of potential binary interactions might pose challenges that the current theoretical framework does not address.

## Recommendations
- Strengthen the theoretical analysis by providing formal guarantees or bounds on the approximation quality of FIGS-BD.
- Include a comprehensive theoretical comparison with other interpretable distillation methods for CBMs to better position the contribution.

While the paper presents a practically useful approach with promising empirical results, its theoretical foundations are insufficiently developed. The lack of comparative analysis with alternative methods and formal guarantees significantly weakens its theoretical contribution. The above suggest that the paper requires substantial theoretical strengthening before it can make a significant contribution to the field of interpretable machine learning.

---

### Decision · Program_Chairs · 2025-03-04

Accept